# Velvet Antler Production and Hematological Changes in Male Sika Deers Fed with Spent Mushroom Substrate

**DOI:** 10.3390/ani12131689

**Published:** 2022-06-30

**Authors:** Chongshan Yuan, Min Wu, Syed Muhammad Tahir, Xinyuan Chen, Changze Li, Aiwu Zhang, Wenfa Lu

**Affiliations:** College of Animal Science and Technology, Jilin Agricultural University, Changchun 130118, China; 18844146800@163.com (C.Y.); koo331500@163.com (M.W.); muhammad.tahir1011@gmail.com (S.M.T.); chenxy9797@163.com (X.C.); mbconfuse@163.com (C.L.)

**Keywords:** male sika deer, spent mushroom substrate, apparent digestibility, hematological, velvet antler

## Abstract

**Simple Summary:**

Velvet antler from sika (and other deer species) is a highly valued nutraceutical in traditional Chinese medicine, a market of hundreds of million dollars. Thus, in this case, and in deer breeding in general, reducing the feeding cost of male sika deer is important to improving the profitability of breeding for velvet antlers. Spent mushroom substrate (SMS) is a waste culture medium for mushrooms. The improper handling of SMS causes environmental pollution. Since SMS can be digested and absorbed by ruminants such as sika deer, replacing SMS with a sika deer concentrate supplement can fully utilize the waste resource of SMS and reduce the production cost of velvet antler.

**Abstract:**

At present, spent mushroom substrate (SMS) is a waste resource that is producing a pollution problem in China, and which has some use as animal feed or fertilizer, has not been assessed as a feed for deer. The purpose of this study is to expand the feed of male sika deer and reduce the feeding cost by using the waste resource of SMS. The 10% concentrated supplement was replaced with SMS and the feed intake, apparent digestibility, blood index and velvet production of male sika deer were measured. As the results showed, compared to the control group, the substitution of SMS for 10% of the concentrate supplement decreased the concentration of IgA (*p* < 0.01), replacing 10% concentrated supplement with SMS of Pleurotus ostreatus (SMS-MP) reduced the intake of organic matter (OMI) and improved the digestibility of ether extract (EE), while replacing 10% concentrated supplement with SMS of Flammulina velutipes (SMS-MF) had no effect on apparent nutrient digestibility, feed intake, velvet antler production, and biochemical indexes. In conclusion, SMS had no effect on serum biochemical indexes and the ratio of the feed weight of the deer supplement to the weight of velvet antler (*p* > 0.05). At the same time, SMS could reduce the feed consumption and improve the economy by using SMS as a waste resource.

## 1. Introduction

China has a huge mushroom industry, accounting for about 75% of the world’s annual output [1]. In China, each kilogram of mushroom generates about 5 kg of waste, with a large number of spent mushroom substrate (SMS) being produced through mushroom cultivation, with an annual output of more than 13 million tons [2]. At present, there is no effective disposal method for SMS, which leads to serious environmental problems and wastes this precious resource. With the emphasis on this issue, SMS gradually plays an increasingly important role in agriculture, such as accelerating compost maturation [3], enhancing the morphological growth of pepper and tomato seedlings [4,5], and enhancing soil organic matter and nutrient content [6]. Even though a large amount of SMS has been used, the quantity generated still exceeds that being utilized [7]. Considering the current global environmental protection and economic development, the full utilization of waste SMS in livestock production has become a growing concern. At present, much research has already been conducted on the processing and usage of SMS [8,9]. SMS is considered to be an animal feed with nutritional value, rich in polysaccharides, vitamins, and trace mineral elements [10], which can be easily digested and absorbed by ruminants such as cattle [11] and sheep [12]. We speculate that the sika deer, which is a ruminant animal, could also digest SMS better.

As a ruminant, sika deer is widely bred in the Jilin Province of China, which is mainly used to produce sika deer velvet antler. Velvet antler is the animal nutraceutical most important in traditional Chinese medicine, which is widely used by two billion people in China, Korea, Japan, and some Southeast Asian countries. Several therapeutical effects have been attributed to velvet antler for over 2000 years [13]. Among the latest confirmed effects are the anti-cancer effects both in cell cultures and animal models [14,15,16], and recently developed treatments to revert aging using cell components of velvet antlers [17]. Therefore, it is important to improve the economic benefits of sika deer velvet antler by reducing the feeding cost of male sika deer. Our previous study showed that a quantity of 10% and 20% with SMS of Pleurotus ostreatus (SMS-MP) can safely replace the concentrate supplement of growing sika deer [18]. Therefore, the reasonable addition of SMS to concentrate supplement is the key to improving the economic benefits of sika deer.

The purpose of this study was to evaluate the use of different types of SMS such as SMS-MP and SMS of Flammulina velutipes (SMS-MF) to replace 10% of concentrate supplements. By detecting the nutrient digestibility, velvet antler production, and blood biochemical parameters of male sika deer, we aimed to clarify the safety of different types of SMS in replacing concentrate supplements, in order to reduce the feeding cost and avoid the waste of resources caused by the random disposal of SMS.

## 2. Materials and Methods

### 2.1. Animals, Diets, and Experimental Designs

The trial lasted for 60 days, 30 healthy 3-year-old male sika deer with an average body weight of 93 kg were randomly divided into 3 groups (10 deer/group), living in the artificial feedlot with no grass, tree canopy cover, etc. Sika deer were offered the same level of nutrients except for different types of dried SMS during velvet antler growing stage. The ratio of concentrate supplement to silage was about 4:1. In the current study, SMS-MP and SMS-MF were used to replace 10% of the concentrate supplement in group MP and group MF, respectively, while the control group (group CON) was fed normally. The feed was restricted and offered 3 times per day at 4:30, 10:30, 16:30, and water ad labium. The daily intake of concentrate supplement was limited to 2 Kg/deer. Silage consumption was collected during the last 8 days of the experimental period. The composition of the concentrate supplement is listed in Table 1.

### 2.2. Sample Collection

#### 2.2.1. Samples of Silage, Concentrate Supplements, and Faeces

Feed and fecal samples were collected 1 h after feeding. Silage, concentrate supplements, and fecal weights were recorded for male sika deer daily for the last 8 days of the experiment. Subsequently, the samples were dried to constant weight in a forced-air oven at 65 °C, then the dried samples were pulverized in a Wiley mill to pass through a 2 mm screen and preserved until analysis.

#### 2.2.2. Blood Sampling

On the last day of the experiment, male sika deer were anesthetized with 1 mL of xylazine hydrochloride. Then, 5 mL blood of velvet antler was collected with whole blood vacutainer after taking off deer velvet antler. Blood was centrifuged for 15 min at 3000 rpm to obtain serum, which was used to measure biochemical and immune globulin. Immediately after blood collection, male sika deer were awakened by injecting 1 mL of niclosamide. Serum samples were preserved at −20 °C for further analysis.

#### 2.2.3. Velvet Antler Sampling

Velvet antler sample collection was performed as described previously [19]. Velvet antlers were removed by a professional technician under the guidance of the institutional velveting regime. The procedures were as follows: male sika deer were anesthetized with 1 mL of xylazine hydrochloride. After the deer is anesthetized, wait for 4 min, test the effect of anesthesia, use a bandage to tighten the base of the pedicle, and remove the velvet antlers below the cutting point, and use grass ash to stop the bleeding after removing the velvet antlers. Finally, measure the yield of fresh velvet antlers.

### 2.3. Chemical Analysis

#### 2.3.1. Feed Intake and Nutrient Digestibility

Feed intake was calculated from the difference in silage and concentrate supplement weights before and after feeding. Silage, concentrated supplement, and fecal samples were collected daily for the last eight days of the experiment.

Concentrated supplement, silage, and fecal samples were dried to constant weight in a forced-air oven at 105 °C, the difference in weight before and after drying was recorded, and the dry matter (DM) content was calculated. The N content was determined using a LECO FP-528N/Protein Tester (LECO, Corporation), and the crude protein (CP) content in the sample was calculated by the following equation: CP = N × 6.25. Inject 80 mL of petroleum ether into the Soxhlet fat extractor, 70 °C water for 16 hr, and finally calculate the ether extract (EE) content. Put the sample into a muffle furnace at 525 °C and heat it for 12 h to determine the content of crude ash and organic matter (OM). Energy of SMS and concentrate supplement samples were measured in duplicate using bomb calorimetry (Model 6050, Parr Instrument Company, Moline, IL, USA) utilizing benzoic acid as a standard. Samples were digested with nitric and perchloric acids, and Calcium (Ca) and Phosphorus (P) contents were detected on an inductively coupled plasma mass spectrometer (Varian Inc., Palo Alto, CA, USA).

Due to the limitations of the experimental conditions, all fecal samples could not be weighed, therefore, acid-insoluble ash in 2 N HCL was used as an internal digestibility marker to calculate nutrient digestibility with the following equation:(1)$$N=100-100\times \frac{D}{F}\times \frac{D_{1}}{F_{1}}$$

*N* represents apparent digestibility of nutrients; *D* is the content of indicator in the diet; *F* is the content of indicator in the feces; *F*_1_ is the content of nutrition in the feces; *D*_1_ is the content of nutrition in the diet.

#### 2.3.2. Production of Velvet Antler

Yield of velvet antler (YVA): All velvet antlers removed at the same stage of growth were weighed.

Feed intake/yield of Velvet antler (F/Y): It is the ratio of the feed intake (g) to velvet antler weight (g) for two months.

#### 2.3.3. Serum Biochemical Indexes

The concentration of albumin (ALB), total protein (TP), globulin (GLO), calcium (Ca), glucose (GLU), urea nitrogen (BUN), inorganic phosphorus (IP), amylase (AMY), cholesterol (CHOL), alanine aminotransferase (ALT), total bilirubin (TBIL), alkaline phosphatase (ALP), inosine (CRE) and creatine kinase (CK) in serum were measured by A25 automatic hematology analyzer (Biosystems SA, Barcelona, Catalunya, Spain) according to the manufacturer’s instructions.

#### 2.3.4. Immune Globulin

Enzyme-linked immunoassay kit (Shanghai Enzyme Biotechnology Co., Ltd., Shanghai, China) was used to detect immunoglobulin A (IgA), immunoglobulin G (IgG), and immunoglobulin M (IgM) of serum according to the manufacturer’s instructions. Briefly, the sample was added to the enzyme-labeled well, biotin-labeled recognition antigen was added, incubated at 37 °C for 1 h, washed with PBST to remove unbound biotin antigen, and then added avidin-HRP, incubated at 37 °C for 1 h, Add chromogenic solution A, chromogenic solution B and stop solution respectively, detect the absorption peak at 450 nm.

### 2.4. Statistical Analysis

Statistical analysis was via least-squares analysis of variance (ANOVA), following the general linear model’s procedure of SPSS (SPSS 19.0 for Windows; SPSS Inc., Chicago, IL, USA). Each experiment was repeated at least three times. All data are presented as least squares means ± SD, and the data were compared by one-way ANOVA and subsequent Duncan test. *p* < 0.01 was considered as highly significant; *p* < 0.05 was considered as significant.

## 3. Results

### 3.1. Nutrient Composition of Different Dried SMS

In this study, we chose SMS-MP and SMS-MF with corncob as the main component to replace the concentrate supplement of sika deer. Nutrient composition of SMS is presented in Table 2. There was no significant difference in nutritional composition between SMS-MP and SMS-MF (*p* > 0.05).

### 3.2. Dietary Nutritional Ingredients of Dried Concentrate Supplements

The dietary nutritional ingredients of dried concentrate supplements are presented in Table 3. There was no significant difference in the nutrient ingredients of the concentrate supplements between the groups after replacing 10% of the concentrate supplement with SMS-MP and SMS-MF (*p* > 0.05).

### 3.3. Apparent Digestibility

Sika deer fed with SMS-MP decreased digestibility of OM (*p* < 0.05), compared with that of group MF (Table 4). Digestibility of EE was higher in group MP than that of group CON and group MF (*p* < 0.05), and there was no significant difference between group CON and group MF (*p* > 0.05).

### 3.4. Feed Intake and Velvet Antler Production

Sika deer fed with SMS-MP decreased OMI (*p* < 0.05), compared with that of group CON. YVA of group MF was significantly higher than that of group MP (*p* < 0.05) as indicated in Table 5. No differences were found in F/Y between the groups (*p* > 0.05).

### 3.5. Serum Biochemical Indexes

The effects of the SMS in the diets on serum biochemical indexes are shown in Table 6. It was indicated that GLU content in group MP was higher than that of group MF (*p* < 0.05). As well, there was no significant difference in other serum biochemical indexes among these three groups (*p* > 0.05).

### 3.6. Immune Globulin

Effects of the SMS in the diets on immune globulin were shown in Table 7. Results indicated that IgA content in group CON is greater than those of group MP and MF (*p* < 0.01), and there were no significant differences in IgG and IgM (*p* > 0.05).

## 4. Discussion

As waste, SMS is a cheap feed for ruminants. SMS can reduce the cost of animal feeding and make full use of this waste resource. The present work provides information on the feed intake, apparent digestibility, serum index, and velvet antler yield of male sika deer. As the results showed, different types of SMS can be used as concentrate supplement for male sika deer. The experiment showed that there were no significant differences in the nutritional components of SMS of different mushroom types. At the same time, the nutritional components of concentrate supplement were not changed after being replaced by SMS. There are no public data about the effects of SMS on the digestibility of nutrients in male sika deer. According to this study, compared to the control group, the digestibility of EE was increased by SMS-MP in the diets. Xu et al. [20] found that the increase of SMS in the total mixed diet could decrease the digestibility of DM, OM, and CP. However, the results of this study showed that SMS had no effect on the digestibility of OM, CP, and DM. In addition, compared with SMS-MP, whether SMS-MF could improve the digestibility of OM requires further study. According to previous studies, the addition of SMS to the diet does not affect digestibility in ruminants [11], which is consistent with our findings; 60 days of feeding with SMS did not adversely affect digestibility in male sika deer.

The feed intake of male sika deer is affected by various factors such as health status and environmental factors. Small changes in dietary conditions would reduce feed intake, so sika deer need a stable diet formula [21]. DMI and OMI generally decrease with increasing neutral detergent fiber (NDF) concentration [22]. It has been reported that high NDF diets reduce the number of meals per day in dairy cows and improve rumen fermentation, and plasma metabolites [23]. However, another study reported that feeding different diets with NDF did not affect rumen fermentation, and there was no interaction between dietary NDF concentration and digestibility [24]. In this study, feed intake and apparent digestibility were not affected in animals fed SMS-MF. The reduction of OMI by SMS-MP may be related to the content of NDF, which needs further study. Velvet antlers are renewed once a year and new antlers grow from permanent bony protuberances. It is well-known that amphibian limbs regenerate based on stem cell differentiation. In contrast, deer antler stem cells are present in the periosteum of the pedicle [25]. The protein and energy content in the diet had a great influence on the yield of velvet antler. Compared with SMS-MP, SMS-MF can significantly increase YVA, but overall SMS does not significantly affect YVA, which is consistent with the trend of DMI and OMI. It could be the result of individual differences. SMS-MP can reduce the value of DMI and F/Y, therefore, we can conclude that the use of different types of SMS instead of concentrate supplement did not affect the feed intake and velvet antler production of male sika deer.

When testing the serum biochemical indicators of male sika deer by SMS, we found that compared with SMS-MP, SMS-MF increased serum GLU concentrations. GLU could be released and oxidized during systemic circulation to provide energy and lead to the synthesis of fatty acids [26]. GLU produced by the liver can be converted to pyruvate after being transported to skeletal muscle, and when the availability of GLU in the body is insufficient, animals require gluconeogenesis to produce GLU. At this time, pyruvate can be converted back to GLU in the liver through the process of gluconeogenesis to provide energy for the body, so, it is an important energy source for body activities and is beneficial to sika deer which are almost completely dependent on the continuous energy supply of GLU for maintaining the material metabolism and organ functions of the body [27]. However, GLU did not differ between groups in this study. Analysis of hematological parameters could help to assess the general health of the animal. Serum BUN concentration has an important relationship with nitrogen metabolism, which can reflect the ratio of protein and non-protein nitrogen levels in ruminant diets [28]. High levels of protein in the diet raise serum ammonia levels and disrupt the immune system of dairy cows [29]. As the results showed, the content of BUN tended to decrease in animals fed SMS, but there was no significant difference. It was similar to those obtained by Liu et al. [11]. In the present study, SMS had no effects on ALT, which is in agreement with the findings of Zeng et al. [30]. According to the study, we can conclude that SMS has no effect on biochemical indexes.

IgA, IgG, and IgM are associated with immune functions and processes, including humoral immunity, cellular immunity, protection from bacterial infection, defense against pathogens, and immune homeostasis monitoring [31]. Among them, IgA plays a key role in defending against microbial infections and protecting mucosal surfaces [32]. Certainly, patients with selective IgA deficiency have been known to develop a variety of infectious and autoimmune ailments [33]. In addition, it has been reported that external stress stimuli to the body would increase the synthesis and transport of IgA in the body through the transmission of sympathetic nerves [34]. In our study, IgA levels in male sika deer fed SMS were lower than that of the control group. It seems to indicate that SMS can reduce acute stress caused by cutting velvet antler.

The addition of SMS had no adverse effect on male sika deer, indicating that SMS can be used as sika deer feed for long-term consumption. The present study has limitations to be underscored. SMS-MP-reduced digestibility of EE and OMI needs to be analyzed by further studies. At the same time, we only selected SMS-MP and SMS-MF for experiments, and we could explore whether more types of SMS can also be used as concentrate supplements for male sika deer in the future.

## 5. Conclusions

SMS can be safely used in feed to replace the concentrate supplement. It has no effect on the health and velvet antler production of male sika deer and it can improve the economic benefits of the velvet antler industry. At the same time, it could provide a new method for the utilization of SMS.

## Figures and Tables

**Table 1 animals-12-01689-t001:** Concentrate supplements composition of male deer (%).

Composition	Content
Corn	59.23
Soybean meal	39.76
NaCl	0.76
Limestone meal	0.15
Ca(HCO_3_)_2_	0.10

**Table 2 animals-12-01689-t002:** Nutrient composition of different dried SMS (%, DM).

	SMS-MP	SMS-MF
Crude protein (CP)	3.01 ± 0.82	5.10 ± 0.20
Ether extract (EE)	9.13 ± 0.96	8.53 ± 0.15
Organic matter (OM)	93.23 ± 0.33	89.67 ± 0.39
Energy (MJ/kg)	14.79 ± 1.07	15.67 ± 0.58

**Table 3 animals-12-01689-t003:** Dietary nutritional ingredients of dried concentrate supplements (%, DM).

	CON	MP	MF
Crude protein (CP)	13.36 ± 0.74	13.78 ± 0.74	12.14 ± 0.76
Ether extract (EE)	12.78 ± 0.12	12.28 ± 0.41	13.28 ± 0.21
Organic matter (OM)	96.45 ± 0.02	96.69 ± 0.19	96.42 ± 0.01
Energy (MJ/kg)	15.15 ± 0.25	15.31 ± 0.31	15.54 ± 0.57

**Table 4 animals-12-01689-t004:** Effect of SMS on apparent nutrient digestibility of male sika deer.

	CON	MP	MF
Digestibility of DM (%)	72.93 ± 6.61	66.97 ± 5.45	73.13 ± 15.15
Digestibility of CP (%)	64.36 ± 7.14	66.60 ± 9.78	68.52 ± 18.59
Digestibility of EE (%)	78.71 ± 4.76 ^b^	83.75 ± 2.89 ^a^	77.89 ± 9.82 ^b^
Digestibility of OM (%)	77.91 ± 5.13 ^ab^	73.76 ± 4.38 ^b^	78.47 ± 9.35 ^a^

Note: DM: dry matter; CP: crude protein; EE: ether extract; OM: organic matter. Means for different lowercase letters were significantly different (*p* < 0.05).

**Table 5 animals-12-01689-t005:** Effect of SMS on feed intake and velvet antler production of male sika deer.

	CON	MP	MF
DMI (g·d^−1^)	2816.54 ± 260.94 ^ab^	2737.58 ± 75.95 ^b^	3120.06 ± 180.43 ^a^
OMI (g·d^−1^)	2714.03 ± 251.76 ^A^	2640.68 ± 73.14 ^B^	3005.74 ± 174.05 ^A^
YVA (g/deer)	1917.14 ± 377.42 ^ab^	1878.57 ± 233.47 ^b^	2300.00 ± 439.35 ^a^
F/Y	74.45 ± 16.62	60.29 ± 6.93	68.59 ± 14.72

Note: DMI: Dry matter intake; OMI: Organic matter intake; YVA: yield of velvet antler; F/Y: feed intake/yield of Velvet antler. Means for different lowercase letters were significantly different (*p* < 0.05). Means for different capital letters were extremely significantly different (*p* < 0.01).

**Table 6 animals-12-01689-t006:** Effect of SMS on serum biochemical indexes of male sika deer.

	Abbreviation	CON	MP	MF
Albumin (g/L)	ALB	28.68 ± 1.73	27.90 ± 3.19	28.33 ± 2.47
Total protein (g/L)	TP	68.10 ± 7.07	68.03 ± 3.18	71.68 ± 4.38
Globulin (g/L)	GLO	39.43 ± 7.37	40.13 ± 5.89	43.35 ± 6.48
Calcium (mmol/L)	Ca	2.03 ± 0.21	2.18 ± 0.18	2.00 ± 0.07
Glucose (mmol/L)	GLU	6.15 ± 1.32 ^ab^	7.82 ± 1.09 ^a^	5.71 ± 1.09 ^b^
Urea nitrogen (mmol/L)	BUN	9.05 ± 1.37	6.82 ± 1.09	8.17 ± 2.94
Inorganic phosphorus (mmol/L)	IP	2.80 ± 1.37	2.41 ± 0.28	2.85 ± 0.41
Amylase (U/L)	AMY	8.25 ± 1.50	10.25 ± 6.85	8.25 ± 2.22
Cholesterol (mmol/L)	CHOL	2.67 ± 0.38	2.89 ± 0.53	2.99 ± 0.40
Alanine aminotransferase (U/L)	ALT	68.50 ± 24.46	70.00 ± 37.21	69.00 ± 19.85
Total bilirubin (umol/L)	TBIL	6.84 ± 2.45	7.49 ± 6.98	5.34 ± 1.73
Alkaline phosphatase (U/L)	ALP	623.00 ± 213.15	467.75 ± 226.65	530.75 ± 296.70
Inosine (umol/L)	CRE	90.00 ± 9.20	99.25 ± 11.73	105.00 ± 10.68
Creatine kinase (U/L)	CK	328.50 ± 164.45	396.00 ± 233.85	316.25 ± 105.28

Note: Means with different lowercase letters within a row differ significantly (*p* < 0.05).

**Table 7 animals-12-01689-t007:** Effect of SMS on immune globulin of male sika deer (mg/mL).

	CON	MP	MF
IgA	20.07 ± 2.34 ^A^	18.24 ± 2.21 ^B^	15.24 ± 4.27 ^B^
IgG	28.78 ± 6.34	27.31 ± 9.28	29.81 ± 8.39
IgM	2.52 ± 1.49	3.12 ± 1.77	2.81 ± 2.15

Note: Means with different uppercase letters differ extremely significantly (*p* < 0.01).

## Data Availability

The dataset generated and/or analyzed during the current study is available from the corresponding author on reasonable request. And they will be provided during review.

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
