# Peer review of "Velvet Antler Production and Hematological Changes in Male Sika Deers Fed with Spent Mushroom Substrate"

_animals, 2022, doi:10.3390/ani12131689_

Round 1
Reviewer 1 Report
This manuscript was difficult to read for content because the English was poor. I kept getting distracted by the English and could not focus on the science. I suggest that the authors work with a native English speaker to complete a detailed revision of the text. In addition, it is my opinion that an abstract should have a sentence or two of justification before jumping in the objective. There also needs to be a more clear description of the methods so that the study could easily be repeated. The graphs need to clearly illustrate the points that the authors are wanting the readers to see. The figure legends need to be more descriptive so that they can stand alone without the reader needing to go to the text to understand what is being illustrated. Finally, a single sentence or two sentences does not comprise a paragraph. A paragraph should begin with a topic sentence, have at least one sentence in the body, and have a summary/transition sentence at the end. There are several places throughout the manuscript where this style and flow can be improved. Once all the style and grammar difficulties are addressed, I could more fully evaluate the science.
Author Response
Dear editor,
Thank you for your letter and for the reviewers’ comments concerning our manuscript entitled “Velvet Antler Production and Hematological Changes in Male Sika Deers Fed with Spent Mushroom Substrate”. Those comments are all valuable and very helpful for revising and improving our paper, as well as the important guiding significance to our researches. We have studied the comments carefully and have made detailed correction and supplement which we hope meet with approval. We replied to all the reviewers' concerns point-by-point. The main corrections in the paper and the response to the reviewer’s comments are as following:
Comments from reviewers and Responses:
-Reviewer 1
Comments and Suggestions for Authors
This manuscript was difficult to read for content because the English was poor. I kept getting distracted by the English and could not focus on the science. I suggest that the authors work with a native English speaker to complete a detailed revision of the text. In addition, it is my opinion that an abstract should have a sentence or two of justification before jumping in the objective. There also needs to be a more clear description of the methods so that the study could easily be repeated. The graphs need to clearly illustrate the points that the authors are wanting the readers to see. The figure legends need to be more descriptive so that they can stand alone without the reader needing to go to the text to understand what is being illustrated. Finally, a single sentence or two sentences does not comprise a paragraph. A paragraph should begin with a topic sentence, have at least one sentence in the body, and have a summary/transition sentence at the end. There are several places throughout the manuscript where this style and flow can be improved. Once all the style and grammar difficulties are addressed, I could more fully evaluate the science.
Response: Thanks for your advice. We have made detailed revisions to the text, and we have invited native English speaker Dr. Syed Muhammad Tahir to revise the grammar in the manuscript. We have revised the abstract accordingly “At present, spent mushroom substrate (SMS), as a waste resource, has not been used reasonably, and it is easy to cause environmental pollution.” has been added to the abstract. Materials and methods has been described in detail. We have revised the legends in the manuscript. The writing style in the manuscript has been revised accordingly. Many thanks for your valuable comments on this manuscript.

Reviewer 2 Report
Reviewed manuscript "Antler Production and Hematological Changes in Male Sika Deers Fed with Spent Mushroom Substrate" (animals-1741778) contains the results of very interesting research work of scientific and practical significance. These studies show two benefits of using spent mushroom substrate:
- feeding deer and reduce the production cost of velvet antler
- reducing waste at the same time.
Deer antler cells can proliferate and differentiate rapidly, blood, on the other hand, is a very sensitive indicator of change. This property makes the antler a valuable model for studying potent growth factors.
The experiment was planned properly and carried out on sufficiently numerical material, male Sika deer kept on the professional farm.
Statistical analysis of the obtained results is correct.
Tables and figures presented the results and statistical data were constructed properly.
The discussion was carried out properly and the literature used in this part of the manuscript was chosen accordingly.
However, please describe in more detail how the antlers was collected from the animals.
Line 120- the first time used abbreviations in the text should be expanded as in lines 126 and 127.
Line 82, 125 - sentences should be started with a capital letter.
Line 137 - end the sentence with a full stop.
In summary - the manuscript contains valuable information, issues and after minor revision should be published in Animals.
Author Response
Dear editor,
Thank you for your letter and for the reviewers’ comments concerning our manuscript entitled “Velvet Antler Production and Hematological Changes in Male Sika Deers Fed with Spent Mushroom Substrate”. Those comments are all valuable and very helpful for revising and improving our paper, as well as the important guiding significance to our researches. We have studied the comments carefully and have made detailed correction and supplement which we hope meet with approval. We replied to all the reviewers' concerns point-by-point. The main corrections in the paper and the response to the reviewer’s comments are as following:
Comments from reviewers and Responses:
-Reviewer 2
Comments and Suggestions for Authors
Reviewed manuscript "Antler Production and Hematological Changes in Male Sika Deers Fed with Spent Mushroom Substrate" (animals-1741778) contains the results of very interesting research work of scientific and practical significance. These studies show two benefits of using spent mushroom substrate:
- feeding deer and reduce the production cost of velvet antler
- reducing waste at the same time.
Deer antler cells can proliferate and differentiate rapidly, blood, on the other hand, is a very sensitive indicator of change. This property makes the antler a valuable model for studying potent growth factors.
The experiment was planned properly and carried out on sufficiently numerical material, male Sika deer kept on the professional farm.
Statistical analysis of the obtained results is correct.
Tables and figures presented the results and statistical data were constructed properly.
The discussion was carried out properly and the literature used in this part of the manuscript was chosen accordingly.
However, please describe in more detail how the antlers was collected from the animals.
Response: Thanks for your advice. Velvet antler sample collection detail is as follows: “Velvet antler sample collection was performed as described previously [17]. Velvet antlers were removed by a professional technician under the guidance of the institutional velveting regime. The procedures were as follows: male sika deer were anesthetized with 1 mL of xylazine hydrochloride. after the deer is anesthetized, wait for 4 minutes, test the effect of anesthesia, use a bandage to tighten the base of the pedicle, and remove the antler below the cutting point, and use grass ash to stop the bleeding after removing the antlers. Finally, measure the yield of fresh velvet antlers. ”
[17] Bao K et al., Effects of dietary manganese supplementation on nutrient digestibility and production performance in male sika deer (Cervus Nippon). Anim Sci J 2017;88:463-467.
Line 120- the first time used abbreviations in the text should be expanded as in lines 126 and 127.
Response: Abbreviations have been expanded accordingly as following: albumin (ALB), total protein (TP), globulin (GLO), calcium (Ca), glucose (GLU), urea nitrogen (BUN), inorganic phosphorus (IP), amylase (AMY), cholesterol (CHOL), alanine aminotransferase (ALT), total bilirubin (TBIL), alkaline phosphatase (ALP), inosine (CRE) and creatine kinase (CK)
Line 82, 125 - sentences should be started with a capital letter.
Response: Thanks for your advice. The issues have been resolved.
Line 137 - end the sentence with a full stop.
Response: Thanks for your advice. The problem has been revised.
In summary - the manuscript contains valuable information, issues and after minor revision should be published in Animals.
Response: Thank you for supporting our research.

Reviewer 3 Report
Line 71. Include postprandial hours
Table 1. Include the nutritional value of the diet.
2.4 Include the statistical model of the experiment
The methodology lacks details. There is no description of how feces and feed were collected to calculate digestibility, including postprandial hours of collection during the number of sampling days.
There is no information on the animals, age, weight, sex, etc.
Why was the analysis of NDF and FDA not carried out?, mainly NDF. Fiber analysis is very important in these experiments.
The methodology does not explain how energy was measured, because energy digestibility was not performed?
I think it is better to present the results in tables. The exact data and the standard deviations are not shown in the graph.
Author Response
Dear editor,
Thank you for your letter and for the reviewers’ comments concerning our manuscript entitled “Velvet Antler Production and Hematological Changes in Male Sika Deers Fed with Spent Mushroom Substrate”. Those comments are all valuable and very helpful for revising and improving our paper, as well as the important guiding significance to our researches. We have studied the comments carefully and have made detailed correction and supplement which we hope meet with approval. We replied to all the reviewers' concerns point-by-point. The main corrections in the paper and the response to the reviewer’s comments are as following:
Comments from reviewers and Responses:
-Reviewer 3
Comments and Suggestions for Authors
Line 71. Include postprandial hours
Response: Thanks for your advice. Postprandial hour is 1 hour. Already added in manuscript.
Table 1. Include the nutritional value of the diet.
Response: The nutritional value of the diet has been supplemented in the Results section.
2.4 Include the statistical model of the experiment
Response: The statistical model was added as following : “Statistical analysis was via least-squares analysis of variance (ANOVA), following the general linear models procedure of SPSS (SPSS 19.0 for Windows; SPSS Inc., Chicago, IL, United States). Each experiment was repeated at least three times.”
The methodology lacks details. There is no description of how feces and feed were collected to calculate digestibility, including postprandial hours of collection during the number of sampling days.
Response: Thanks for your advice. We have supplemented the details in the test methods accordingly.
“Feed and fecal samples were collected 1 h after feeding, and silage, concentrate supplements, and fecal weights were recorded for male sika deer daily for the last 8 days of the experiment.”
“acid-insoluble ash was used as an internal digestibility marker to calculate nutrient digestibility”
There is no information on the animals, age, weight, sex, etc.
Response: Thanks for your advice. We have made the following changes: “The trial lasted for 60 days, 30 healthy 3-year-old male sika deer with an average body weight of 93 kg were randomly divided to 3 groups (10 deer/group)”
Why was the analysis of NDF and FDA not carried out?, mainly NDF. Fiber analysis is very important in these experiments.
Response: Thanks for your advice. We considered the importance of NDF and FDA in the design of the trial. When we studied through the literature, we found that the effect of NDF on ruminants was not completely consistent, so we did not carry out related work, but we have supplemented the relevant information on NDF in detail in the manuscript Discussion as following: “DMI and OMI generally decrease with increasing of neutral detergent fiber (NDF) concentration [21]. It has been reported that diets high NDF diet reduce the number of meals per day in dairy cows and improve rumen fermentation, plasma metabolites [22]. However, another study reported that feeding different diets with NDF did not affect rumen fermentation, and there was no interaction between dietary NDF concentration and digestibility [23]. In this study, feed intake and apparent digestibility was not affected in animals fed SMS-MF. The reduction of OMI by SMS-MP may be related to the content of NDF, which needs further study. ”
[21] Goulart RS et al., Effects of source and concentration of neutral detergent fiber from roughage in beef cattle diets on feed intake, ingestive behavior, and ruminal kinetics. J Anim Sci 2020;98.
[22] Cao Y et al., Physically effective neutral detergent fiber improves chewing activity, rumen fermentation, plasma metabolites, and milk production in lactating dairy cows fed a high-concentrate diet. J Dairy Sci 2021;104:5631-5642.
[23] Kendall C, Leonardi C, Hoffman PC, Combs DK, Intake and milk production of cows fed diets that differed in dietary neutral detergent fiber and neutral detergent fiber digestibility. J Dairy Sci 2009;92:313-23.
The methodology does not explain how energy was measured, because energy digestibility was not performed?
Response: The energy measurement method has been added to the Materials and Methods: “Energy of SMS and concentrate supplements samples was measured in duplicate using bomb calorimetry (Model 6050, Parr Instrument Company, Moline, IL) utilizing benzoic acid as a standard.” When we tested the organic matter digestibility results, we found that compared with the control group, there was no significant difference in the organic matter digestibility between the MP group and the MF group, so we did not continue to do the energy digestibility test.
I think it is better to present the results in tables. The exact data and the standard deviations are not shown in the graph.
Response: Thanks for your advice. We have modified the figures into tables.

Round 2
Reviewer 3 Report
The authors have made the corrections noted by the reviewers. The paper has better presentation and is ready for publication.
Author Response
The authors have made the corrections noted by the reviewers. The paper has better presentation and is ready for publication.
Response: Thanks for your constructive comments on the manuscript.